# Hemostasis Evaluation of Antibacterial and Highly Absorbent Composite Wound Dressings in Animal Hemostasis Models

**DOI:** 10.3390/polym14091764

**Published:** 2022-04-26

**Authors:** Yu-Tung Shih, An-Pang Chen, Mei-Feng Lai, Mei-Chen Lin, Bing-Chiuan Shiu, Ching-Wen Lou, Jia-Horng Lin

**Affiliations:** 1Division of General Neurosurgery, Jen-Ai Hospital, Dali District, Taichung City 412224, Taiwan; shihyutung@gmail.com; 2Technical Center, Fujian Changyuan Textile Co., Ltd., Fuzhou 350200, China; duncannano@gmail.com; 3Laboratory of Fiber Application and Manufacturing, Department of Fiber and Composite Materials, Feng Chia University, Taichung City 40724, Taiwan; ritalin2870@gmail.com; 4College of Material and Chemical Engineering, Minjiang University, Fuzhou 350108, China; toyysbk2@yahoo.com.tw; 5Department of Bioinformatics and Medical Engineering, Asia University, Taichung City 413305, Taiwan; 6Department of Medical Research, China Medical University Hospital, China Medical University, Taichung City 404333, Taiwan; 7Advanced Medical Care and Protection Technology Research Center, College of Textile and Clothing, Qingdao University, Qingdao 266071, China; 8Fujian Key Laboratory of Novel Functional Fibers and Materials, Minjiang University, Fuzhou 350108, China; 9Innovation Platform of Intelligent and Energy-Saving Textiles, School of Textile Science and Engineering, Tiangong University, Tianjin 300387, China; 10School of Chinese Medicine, China Medical University, Taichung City 404333, Taiwan; 11Advanced Medical Care and Protection Technology Research Center, Department of Fiber and Composite Materials, Feng Chia University, Taichung City 407102, Taiwan

**Keywords:** nanoparticles, Tencel^®^ fiber, high absorption fibers, antibacterial, hemostasis, composite dressings

## Abstract

To reduce the bleeding time and to shorten the surgery time are vital to patients’ prog-nosis, therefore, in this study, high moisture absorption nonwoven composites are proposed to attain hemostasis in time. Polyacrylate fiber and Tencel^®^ fibers at different blending ratios (10:90, 20:80, 30:70, 40:60, and 50:50) are used to form PT composite nonwoven. Next, composed of a 50:50 ratio, PT composite nonwoven exhibits the maximal vertical wicking height of 4.4 cm along the cross direction. Additionally, the UV-Vis absorption spectra analysis shows that at absorption waves of 413–415 nm, the occurring of distinct peaks suggests the presence of nanoparticles. The XRD patterns indicate the presence of silver nanoparticles with corresponding crystal planes of characteristic peaks at (111), (200), and (220). Polyacrylate/Tencel^®^ nonwoven composites exhibit comparable adsorption capacity of blood and water molecules. In particular, 30PT composite nonwoven outperforms the control group, exhibiting 3.8 times and 4.7 times greater the water absorption and blood absorption, respectively. Moreover, a great number of red blood cells with a size of 4–6 μm agglomerate among fibers as observed in SEM images, while 6hr-PT composite dressing demonstrates the optimal antibacterial efficacy against Escherichia coli and Staphylococcus aureus, proven by the zone of inhibition being 1.9 mm and 0.8 mm separately. When in contact with plasma, hemostasis composites have plasma hemostasis prothrombin time of 97.9%, and activated partial thromboplastin time of 96.7%. As for animal hemostasis model, the arteria over the rats’ thigh bones is cut open perpendicularly, generating mass arteria hemorrhage. To attain hemostasis, it takes 46.5% shorter time when using composite dressings (experimental group) than the control group.

## 1. Introduction

Excessive hemorrhage is a major cause of death in wars, accidents, and surgical treatments, which means that a reduction in either the hemorrhage level or the surgery time has a significant impact over the prognosis of patients. Therefore, hemostasis materials are of the most importance to patients’ survival. Antibacterial efficacy of wound dressings is required because microbes enter wounds and the surrounding environment, propagating and causing inflammation, and as such, jeopardizing the tissue and delaying the healing [1].

The hemostasis materials demand a high moisture absorption capacity because the blood contains considerable water. Traditional hemostasis gauze is mainly ordinary medical gauze; other common hemostasis materials include gelatin sponge [2,3], microcrystalline fibers, and oxidized regenerated cellulose [4,5,6], exhibiting a certain but unsatisfactory hemostasis efficacy. For example, when left inside the patient’s body, gelatin sponge may exacerbate the adhesion with tissues in the surgical site. Subsequently, the development in hypercoagulant materials that can replace traditional gauge has become an emphasis in the biomedical research that aims at investigating new types, structures, and properties of biological materials. In other words, the design and development of the innovative functional materials are expected to enhance the biocompatibility [7] and blood coagulation.

Acrylate fibers are superabsorbent polymers that are made of propylene via polymerization. In the meanwhile, polyacrylate fibers have been pervasively investigated, and then proved to be safe and toxicity-free for the applications regarding allergy, inflammation, dermatosis, gene, reproduction, and digestion [8]. Environment-friendly fibers are also commonly used as hemostatic materials otherwise. For example, lyocell fibers are brand new fibers that are produced via spinning wood pulp with N-Methylmorpholine oxide as a solvent, and the production process does not discharge harmful substances to jeopardize the environment. In fact, 99.5% of the spinning solvents can be recycled, which advances the resulting fibers to being innovative and eco-friendly [9,10,11,12]. As a result, lyocell fibers have high moisture absorption and good structural stability, so they qualify as hemostatic materials [13].

Chitosan and cellulose share comparable molecular structures where molecules are in the form of straight chains. In addition to strong polarity and ease of crystallization, chitosan has a melting point that is higher than the self-decomposition temperature, which makes it difficult to acquire amorphous chitosan. In the medical applications, chitosan helps accelerate platelet aggregation, and has low toxicity, biodegradable property, and good biocompatibility [14,15]. Furthermore, chitosan facilitates coagulation because it contains cationic properties that aggregate red blood cells and platelets [16]. Subsequently, the clots are generated when the blood contacts chitosan. Plasma proteins, including albumin, γ—globulin, blood fibrinogen, and prothrombin, are efficiently adsorbed over the heterogeneous materials, namely, plasma proteins introduce platelets to adhere to the surface of materials [17,18,19].

In this study, Tencel^®^ fibers and polyacrylate fibers are made into polyacrylate/Tencel^®^ (PT) nonwoven composites, and the PT nonwoven composites are evaluated in terms of physical evaluations as related to the fiber blending ratio, thereby acquisition of the optimal moist absorption for PT nonwoven composites. Ag nanoparticles are added during the manufacturing process of PT nonwoven composite in order to examine the antimicrobial activity. Furthermore, FTIR, UV-Vis absorbent Spectra, and X-ray diffraction are employed to investigate the formation of nano Ag particles. Lastly, PT nonwoven composites with extracorporeal blood absorption as well as hemostasis in animal models are obtained and examined with the clinical hemostasis efficacy for clinical use. Afterwards, health care workers are able to select suitable hemostasis materials according to the bleeding status.

## 2. Materials and Methods

### 2.1. Materials

Tencel^®^ staple fibers (Taiwan Web-Pro Co., Ltd., Kaohsiung, Taiwan) have a length of 51 mm and a fineness of 1.7 D. Polyacrylate fibers (Asiatic Fiber Corporation, Taoyuan City, Taiwan) have a length of 48 mm and a fineness of 9 D. Polyacrylate fiber (HAF) were purchased from Asiatic Fiber Corp., Taiwan. Chitosan powders at 80% degree of deacetylation were procured from Global Biological Technology Co., Ltd., Kaohsiung City, Taiwan. Silver nitrate solution (extra pure grade) was provided by Union Chemical Works Ltd., Kaohsiung City, Taiwan.

### 2.2. Preparation of Polyacrylate/Tencel^®^ (PT) Nonwoven Composites

Through opening, blending, carding, lamination, and needle punching, polyacrylate fiber was processed into PT nonwoven composites with different polyacrylate fiber contents (10, 20, 30, 40, and 50 wt %) and needle punching density of 100 punches/cm^2^. The composites have a basis weight of 100 g/m^2^. Schematic illustration of the PT nonwoven composites’ manufacturing process is shown in Figure 1.

In addition, water accounts for 2/3 of blood, and with the majority of water is absorbed blood which has a higher viscosity, which in turn facilitates the coagulation of platelets and eventually achieves hemostasis. Finally, PT nonwoven composites are evaluated with moisture absorption evaluation and vertical wicking height assessment to determine the optimal parameters. The parameters of PT nonwoven composites was show in Table 1.

### 2.3. Preparation of Nano Ag-Chitosan Composite Dressing

Chitosan powders are dissolved in acetic acid to formulate chitosan solution at a concentration of 2 wt %. Silver nitrate is then added to improve the antibacterial properties of chitosan, during which the reaction time is changed as 2, 4, or 6 h, thereby examining the influence of reaction time on the properties of Ag-chitosan mixtures. The reaction process is referred to in a previous study [20]. The mixtures are evaluated with UV/Vis spectrometer, Fourier transform infrared spectroscopy analysis, and X-ray diffraction analysis, thereby yielding the optimal Ag-chitosan mixture. Next, the mixture is used to coat nonwoven composites, after which freeze-drying is employed. The resulting nano Ag-chitosan composite dressings are evaluated with PT and APPT analyses, blood absorption measurement, and animal hemostasis models, thereby determining the optimal nano Ag-chitosan composite dressing.

### 2.4. Tensile Strength Test

The tensile strength and elongation at break of nonwoven matrices, composite matrices, and nonwoven composites are measured at a rate of 305 ± 13 mm/min using a universal strength testing machine (Instron 5566, Norwood, MA, USA) as specified in ASTM D5035-06 Standard Test Method for Breaking Force and Elongation of Textile Fabrics (Strip Method). The distance between the gauges is 76 mm. Samples have a size of 180 mm × 25.4 mm. Five pieces for each specification are taken along the machine direction (MD) and the cross machine direction (CD). MD is the discharge direction from the machine while CD is the direction that is perpendicular to MD. The test is conducted as related to each parameter with an attempt to determine optimal parameters. The data are then averaged to compute standard deviation and coefficient of variation.

### 2.5. X-ray Diffractometer (XRD) Analysis

An XRD (36 MXP3, Mac Science, Yokohama, Japan) is used to measure Ag-chitosan films for qualitative analysis. The target is made of copper; the wavelength is 0.154 nm; the operation voltage and current is separately 40 kV and 30 mA; the scanning rate is 1°/min, and 2θ has a range of 1–60°. Each compound is composed of different atomic and crystalline structures and lattice constants, so the distance between atomic plane, d (h, k, l), can be used to analyze the components.

### 2.6. Fourier Transform Infrared Spectroscopy (FTIR)

Fourier transform infrared spectroscope (IR Affinity-1, Kyoto, Japan) is used to measure the specified characteristic peaks of Ag-chitosan films. Samples are exposed to infrared radiation, after which the resonance absorption of samples are measured and recorded as related to far infrared rays (FIR) at different wavelengths. Afterwards, the functional group that the sample contains can be surmised based on the specified wavelength that is absorbed by FIR.

### 2.7. Antibacterial Measurement

The antibacterial performance of fabric samples are evaluated as specified in JIS L1902 2002 test standard, using the quantification method. Two types of bacteria strains, including Escherichia coli (ATCC25922) and Staphylococcus aureus (ATCC25923), are made into bacterial suspensions that are cultured in an incubator at 37 °C for 16–18 h, and the concentration is adjusted to 1 × 108 using an UV-Vis spectrometer. Next, 100 μL of a culture medium is used to smear the solid culture tray, after which a ø28 mm circular sample is placed on top of it for a subsequent incubator culture at 37 °C for 16–18 h. Finally, the zone of inhibition of each sample is observed, compared, and analyzed.

### 2.8. Ag Release Rate of PT Composite Dressing

The Ag release rate of samples is determined using the atomic absorption spectroscopy. Weighing 30 mg, PT composite dressing is placed in a flask containing 50 mL of simulated bodily fluid, and then cultured in a shaking incubator at 37 °C for a specified time (1 to 36 h). The atomic absorption spectroscopy is used to examine the Ag release rate of samples as related to the time.

### 2.9. SEM Surface Observation

The surface is observed using a scanning electron microscope (Hitachi S3000N, Tokyo, Japan) where the electro gun generates electron beams with a voltage acceleration of 0.2–40 KV. Usually, the electron beams pass through an electronic optical system consisting of three electromagnetic lenses, and then are converged to irradiate over the surface of sample. The interaction between the converged beam and the sample excite the secondary electrons and reflective electrons, thereby forming the image for observation.

### 2.10. Prothrombin Time (PT) and Activated Partial Thromboplastin Time (APTT)

The blood test in this work was approved by the Institutional Animal Care and Use Committee (IACUC) with Approval No. 100-CTUST-22, and within a protocol period from 1 June 2012 to 5 May 2015. Prothrombin time (PT) and activated partial thromboplastin time (APTT) were measured to determine the effect of wound dressing on blood-clotting time. The PT test was performed on heparinized plasma, and the PT reagent comprised tissue factor, CaCl_2_, and phospholipid. The APTT test was performed by adding APTT reagent into heparinized plasma. The APTT reagent was composed of plasma activator, phospholipid, and CaCl_2_. The PT agent (APTT reagent) and wound dressing sample were added to plasma in the test tube, and PT (APTT) was tested at 37 °C simultaneously.

### 2.11. Animal Test

Rats (Sprague-Dawley, SD) are the subjects in the hemostasis test. For the starter, the rats are inflicted with a wound over a rear limb intentionally. The muscle tissues are cut along the groin area to damage an artery vessel, generating hemorrhage. Next, the wounds are then covered with hemostasis nonwoven composites (the experimental group) or cotton fabrics (the control group) and then pressed for two seconds. The length of time when hemostasis occurs is recorded since the artery vessel was cut.

### 2.12. Statistical Analyses

One-way ANOVA is used in this study. It is a statistical mode of a single analytical response variable to a single categorical explanatory variable, and is used to examine whether there is a significant difference between two or more parameter averages. As for the variance analyses, *p* value indicates the statistical significance of independent variables with corresponding random change, and is represented by * when *p* < 0.05 and ** when *p* < 0.01.

## 3. Results and Discussion

### 3.1. Effects of Polyacrylate Fiber Content and Fiber Orientation on Tensile Strength of PT Nonwoven Composites

PT nonwoven composites are made of polyacrylate and Tencel^®^ fibers. Figure 2 and Figure 3 shows the tensile strength and tensile strain-stress curves of the composites as related to the polyacrylate fiber content and fiber orientation. When the polyacrylate fibers are increased from 10 wt % to 50 wt %, PT nonwoven composites exhibit a decreasing maximal tensile strength. Tencel^®^ fibers have higher mono fiber strength than polyacrylate fibers, which means that a greater content of polyacrylate fibers adversely affects the maximal tensile strength at break. As far as the fiber orientation is concerned, PT nonwoven composites show two times greater maximal tensile strength at break along the CD than that along the MD. As the nonwoven manufacturing process involves opening, mixing, and carding, the majority of fibers are aligned along the carding direction, and at the same time, are leaving CD with a higher fiber orientation in comparison to MD.

### 3.2. Effects of Polyacrylate Fiber Content and Fiber Orientation (MD/CD) on Vertical Wicking Length of PT Nonwoven Composites

Figure 4 shows the correlation between vertical wicking length of PT nonwoven composites, and polyacrylate fiber content and fiber orientation. The composites are composed of 10, 20, 30, 40, or 50 wt % of polyacrylate fibers, and have a basis weight of 100 g/m^2^. The wicking length is in direct proportion to polyacrylate fiber content. As polyacrylate fibers are chemically composed of polyacrylate that swells in multiple times efficiently when in contact with water, water molecules are absorbed by the interior fibers, and samples demonstrated a higher wicking effect. Water molecules are also transferred to fibers next to polyacrylate fibers concurrently, which can be summarized that a greater amount of polyacrylate fibers contribute to a higher vertical wicking length.

Figure 5 shows the fiber distribution of fiber webs that are made of Tencel^®^ and polyacrylate fibers via opening, mixing, and carding procedures. The nonwoven composites are scanned in order to examine the fiber orientation in the angles of 0 to 180 degrees, thereby determining that the majority of fibers are aligned along the CD. In addition, the absorption process and fiber alignment are in a similar direction, which enable fibers to exhibit a maximal capillary phenomenon that rises the vertical wicking length.

Figure 6 shows the hygroscopic curves (vertical wicking length) of PT nonwoven fabrics as related to the content of constituent polyacrylate fibers (0 and 50 wt %). The curves are highly dependent on the composition of nonwoven fabrics. In this study, nonwoven fabrics are used and they have a higher porosity than knitted fabrics and woven fabrics. As a result, nonwoven composites can absorb a considerable amount of water in a short time. It was also found that PT nonwoven composites show efficient adsorption in 30 s regardless of whether it is along the MD or CD. The results are ascribed to the facts that the constituent fibers possess a larger specific surface area and effective capillary action, as well as the porous structure of nonwoven composites demonstrating good water retention characteristics. Subsequently, water can be efficiently transmitted to other fibers that are comparatively drier, and as such, to achieve a greater absorption level. With 0 wt % of polyacrylate fibers, the hygroscopic curves reach a stable state when the nonwoven composites are in contact with water for 3 min. By contrast, with 50 wt % of polyacrylate fibers, the hygroscopic curves reach a stable state when nonwoven composites are in contact with water for 7 min. The comparison shows that a greater content of polyacrylate fibers provides nonwoven composites with better swelling and hygroscopic level.

### 3.3. FTIR Analyses of Chitosan/Ag Membranes

The special interaction between chitosan and metal surface can be analyzed by FTIR. In this study, the FTIR spectrum of pure chitosan and chitosan/Ag membranes is analyzed at a wave frequency range of 650–4000 cm^−1^. Figure 7 shows that the stretching vibrations of N-H and O-H groups may be overlapped, and the higher spectra between 3300–3500 cm^−1^ is N-H group. In the meanwhile, there is a slight shift at 3308.1 cm^−1^, the vibration of which is caused by Ag. The amino group vibration of chitosan occurs at 1570.1 cm^−1^ and then shifts to 1554.7 cm^−1^; however, the O-H group bending vibration at 1417.7 cm^−1^ does not shift distinctively. The test results prove that the interaction between amino group and amido group occurs, so Ag-NPs is amino-cladded.

### 3.4. Effects of Reaction Time on UV-Vis Absorbent Spectra of Ag Nanoparticles

Figure 8 shows the correlation of reaction time and the wavelength with UV-Vis absorption spectra of Ag nanoparticles in an incident wave range of 250–650 nm. The chitosan solution does not exhibit significant absorption peak. Additionally, after silver nitrate is added to the solution, the shade of color in the reactor alters from yellow to brown [21]. Concurrently, there are also distinct absorption waves at 413–415 nm due to the presence of nanoparticles, which is similar to the finding of the study by Naik et al. [22,23]. The growth of Ag nanoparticles is dependent on the surface plasmon resonance [24]. Namely, the surface plasmon occurs over the nanoparticles. When nanoparticles are exposed to incident lights, the conducting electron cloud of nanoparticles are affected by the incident electric field, resulting in a large-scale oscillation that is called the particle plasmon, as seen in Figure 9.

### 3.5. Effects of Chitosan Concentration and Reaction Time on X-ray Diffraction Spectrum of Chitosan/Ag Membranes

Figure 10 shows that with 2θ being 21.9, chitosan/Ag membranes exhibit distinct crystallization characteristic peaks, which is due to the constituent cellulose of chitosan. The two peaks correspond to the α-type and β-type [25], which is similar to the findings of a study by Islama et al. [26]. Consisting of Ag particles, chitosan/Ag membranes show a reaction time of 2 h, and the Miller index is (111) when 2θ is 37.3. With the constantly increasing time, the Miller index is (200) and (220) for 2θ being 43.2 and 63.2, respectively, which indicates the generation of Ag nanoparticles. Especially in a long reaction time, the growth of Ag particles is even more distinct [27]. Due to different lengths of reaction time, the particles may exhibit slightly different nanoscales, which in turn shifts diffraction peaks marginally. However, it is observed that the diffraction peaks belong to face-centered cubic crystals which mean standard Ag crystals.

### 3.6. Water and Blood Absorption of PT Nonwoven Composites

Plasma comprises 55% of the blood, and water comprises 92% of plasma [28]. Therefore, a great moisture absorption is required by hemostasis fabrics, thereby to absorb the massive blood volume and tissue fluid oozed from the wounds. Based on the moisture absorption evaluation, PT nonwoven composites can effectively absorb in 30 s, but the absorption amount needs further investigation. This study compares different PT nonwoven composites in terms of the difference in the absorption capacity of water molecules and blood. Figure 11 shows the water and blood absorption capacity of cotton fabrics (control group) and PT nonwoven fabrics composed of 30 wt % and 50 wt % polyacrylate fibers. Comparing to the control group, 30PT and 50PT composite nonwoven separately exhibits 3.8 times and 4.4 times greater water absorption as well as 4.7 times and 5.2 times greater blood absorption.

Moreover, PT nonwoven fabrics show comparable absorption capacity regardless of whether it is blood or water molecules, which is pertinent to the composition of blood. Water accounts for 70% of blood and therefore, PT nonwoven composites have comparable absorption efficacy of both water and blood. In Figure 12a,b, the SEM images show the blood-absorbed cotton fabrics and 50 PT nonwoven composites where a tremendous amount of red blood cells (4–6 μm) are aggregated among fibers which can be distinguished based on the appearance of fibers. Figure 12d indicates that the red blood cells are adsorbed over the fibers and the morphology of the former is changed concurrently. This phenomenon can be interpreted according to the cellular morphology, and is correlated with the presence of fibrin surrounding red blood cells.

### 3.7. Antibacterial Property of PT Nonwoven Composites

Table 2 shows the antibacterial performances of PT nonwoven dressing against Escherichia coli and Staphylococcus aureus as related to the reaction time. According to the test results, PT nonwoven dressings show antibacterial zone against both Gram-positive and Gram-negative bacteria, especially a reaction time of six hours, because Ag nanoparticles can penetrate the cell walls, killing the bacteria [29]. In addition to a nanoscale, Ag nanoparticles also possess large specific area and volume ratio, and thus can improve the permeability of cell walls, generating active oxygen. In the meanwhile, they release Ag ions to block the regeration of DNA [30]. Based on the theory of Jones and Hoek [31], the most common mode is that the free silver ions are absorbed to interrupt the ATP molecular chains, preventing the formation of DNA or active oxygen [32].

### 3.8. Ag Release Rate of PT Nonwoven Dressing

PT nonwoven dressing is soaked in simulated bodily fluid for a 2 to 8 h incubation at a constant temperature (37 °C), and the Ag release rate is then measured. Figure 13 shows that the test results prove that, regardless of whether the culture time being 2, 4, or 6 h, samples exhibit an efficient Ag release rate. To sum up, the test results indicate that in 24 hours, Ag release rate spikes to the maximal value, especially the 6 h group which demonstrates the highest amount of Ag nanoparticles, namely the maximal Ag release rate. Pawena et al. [33] investigated the release level of silver nanoparticles from fabrics blending cotton, polyester, and cotton fibers. Despite 20 wash cycles, the fabrics still released 48–72% of Ag nanoparticles.

### 3.9. PT, APTT, and Platelet Agglutination Analyses of PT Nonwoven Composites

After the blood separation and before the agglutination reaction, the plasma is at a fluid state as shown in Figure 14a. When rendered with activation, plasma exhibits the agglutination reaction as shown in Figure 14b. On the other hand, PT and APTT are extended on purpose, for which heparin is thus added to plasma in order to prolong the agglutination reaction span. Figure 15 shows that plasma has a PT being 15.2 s and an APTT being 59.8 s originally, the data of which serves as the basis (100%). Moreover, after contacting polyacrylate/Tencel^®^ nonwoven composites, the PT and APTT of plasma are 97.9% and 96.7% individually. The shortened gap may be attributed to the activation between platelets and chitosan, demonstrated by the facts as follows. Platelets grow pseudopodia that then extends and deforms; the integrin composites are activated, and the initiation of calcium ion signals causes adhesion of platelets as well as the speedy bonding with blood fibrin monomers. However, there is no significant impact on the effectiveness because only PT exceeding 3 s than that of the control group accounts for a clinical significance exclusively [34,35].

### 3.10. Hemostasis Evaluation of PT Nonwoven Composites in Animal Hemostasis Model

Figure 16a shows that it takes 380 s for cotton fabrics (the control group) to implement hemostasis. In hospitals or a hemorrhage condition, pure cotton gauze is commonly used to stop bleeding. Pure cotton gauze is composed of 100% of cotton fibers as well as a woven fabric structure via a woven process, serving as the control group in this study. At the same time, the proposed hemostasis nonwoven composites are evaluated for their effectiveness. The test results prove that the proposed materials exhibit efficient moisture absorption while being highly blood absorbent. In the meanwhile, the materials are beneficial for the platelet agglutination in the wounds, achieving blood coagulation. Moreover, the proposed materials contain antimicrobial agents that help hemostasis in animal models efficiently, which may be attributed to cations that chitosan possesses. As a result, chitosan induces the agglomeration of red blood cells and platelets so heterogenous materials can efficiently adsorb plasma protein when they are in contact with blood. In addition, the use of PT nonwoven fabrics also demonstrates hemostasis time of approximate 203 s that is 46.6% shorter than that of the control group. As PT nonwoven composites are made of compound nonwoven fabrics and antimicrobial agents, the marginal decrease in the hemostasis time may be ascribed to chitosan that facilitates blood coagulation. Subsequently, chitosan aggregates platelets to form clots in a bleeding condition. Based on the One-way ANOVA with *p* value being <0.01, there is a significant difference in the hemostasis between cotton nonwoven fabrics and PT nonwoven composites. Wu et al. examined how the collagen sponge type of hemostatic materials healed the bleeding of a rabbit ear and their hemostatic effect over the surface infection wound. The collagen sponge materials exhibited efficient hemostatic effect that reduced the bleeding amount and accelerated the wound healing [36]. Moreover, the hemostatic time was 206 s which is comparable to the results of the proposed material in this study.

## 4. Conclusions

This study successfully combines polyacrylate fibers and Tencel^®^ fibers, forming PT nonwoven composites by means of the nonwoven manufacture. A rise in Tencel^®^ fibers has a positive influence on the tensile strength of PT nonwoven composites. Specifically, with 30 wt % of polyacrylate fibers, PT nonwoven composites show a vertical wicking length of 3.9 cm and moisture absorption rate being 38.1, which are separately 11.4% and 33.9% greater than PT nonwoven composites containing 10 wt % of polyacrylate fibers. In light of the analysis of UV-Vis absorbent spectra of Ag nanoparticles, when UV distinct absorption waves are at 413–415 nm, the peak raises with the reaction time, increasing from two to six hours. The XRD results indicate the generation of Ag nanoparticles, demonstrated by the corresponding Miller index of characteristic peaks at (111), (200), and (220). PT nonwoven composites can absorb similar amount of blood and water. The SEM images demonstrate that a considerable amount of red blood cells agglomerate among fibers.

As for the animal hemostasis model, rats’ arteries’ muscle tissues are cut along the groin area to damage an artery vessel perpendicularly, generating hemorrhage. Comparing to cotton nonwoven fabrics, PT nonwoven composites can save 46.6% of hemostasis time. The experimental results provide worthy development and applications for future wound dressings or hemostasis matrices.

## Figures and Tables

**Figure 1 polymers-14-01764-f001:**
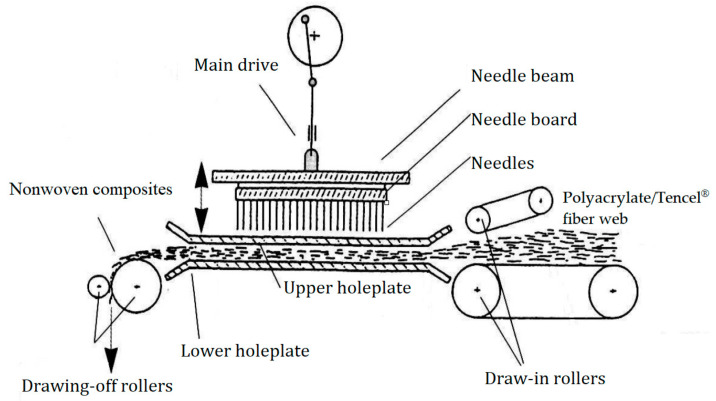
Schematic illustration of the PT nonwoven composites’ manufacturing process.

**Figure 2 polymers-14-01764-f002:**
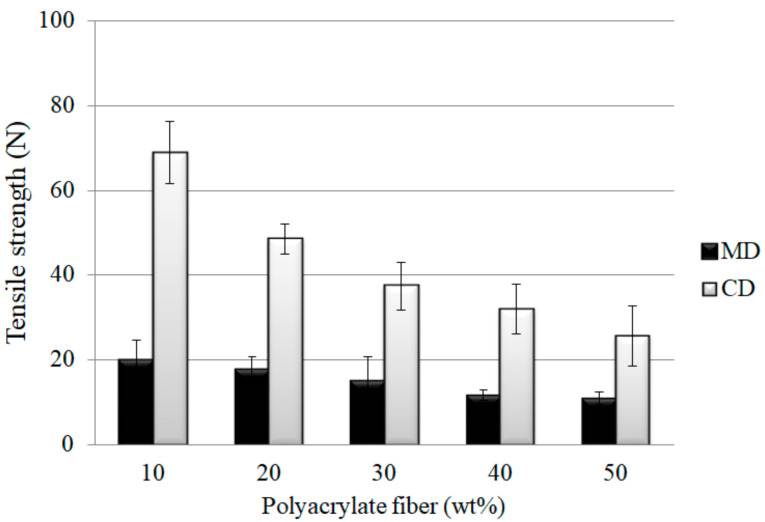
Tensile strength of PT nonwoven composites as related to the content of polyacrylate fibers and fiber orientation.

**Figure 3 polymers-14-01764-f003:**
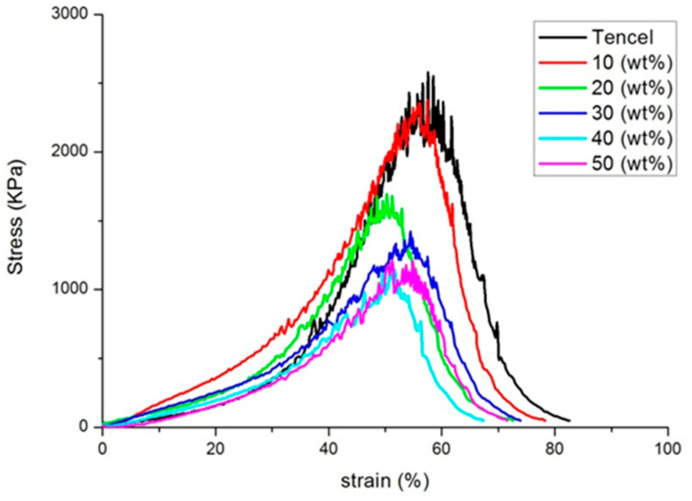
Tensile stress-strain curves of PT nonwoven composites as related to the content of polyacrylate fibers.

**Figure 4 polymers-14-01764-f004:**
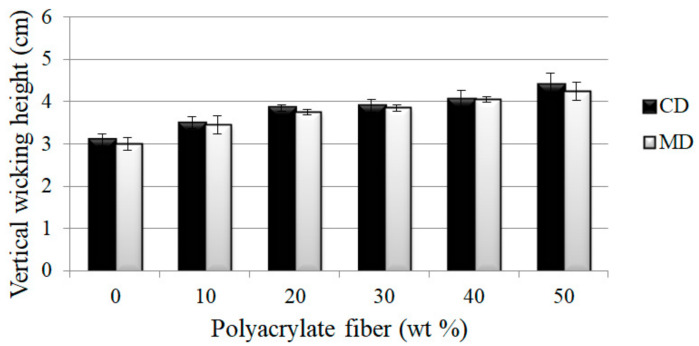
Wicking height of PT nonwoven composites as related to polyacrylate fiber content and fiber orientation (MD, CD).

**Figure 5 polymers-14-01764-f005:**
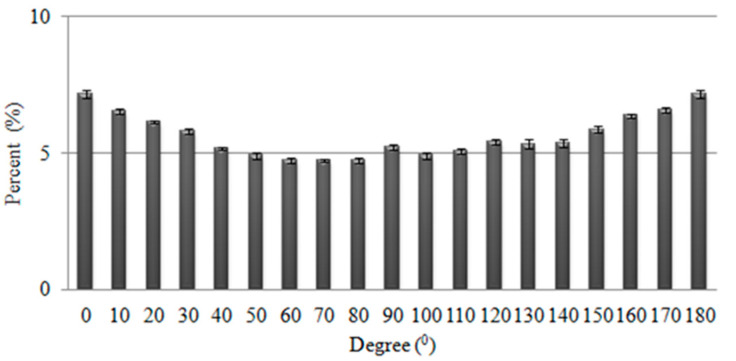
Fiber angle distribution (0–180 degree) of PT nonwoven fabrics.

**Figure 6 polymers-14-01764-f006:**
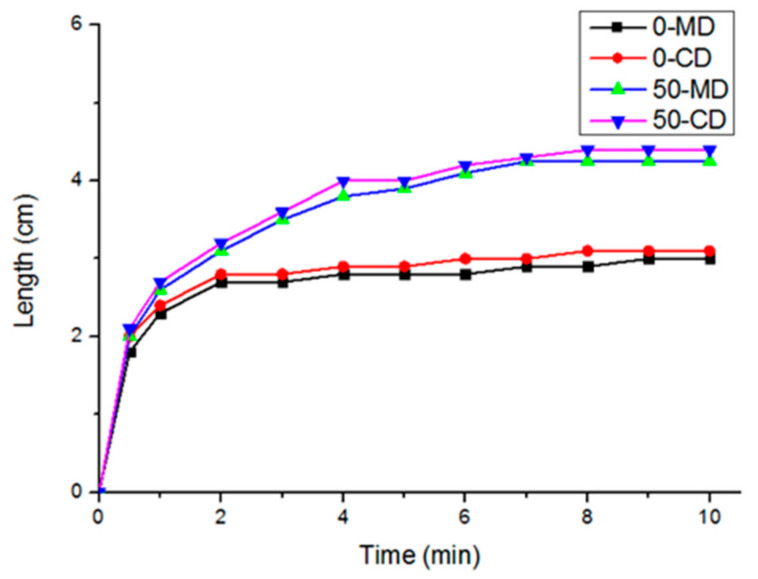
Dynamic curves of PT nonwoven composites as related to polyacrylate fiber content (0 and 50 wt %) and fiber orientation (MD and CD).

**Figure 7 polymers-14-01764-f007:**
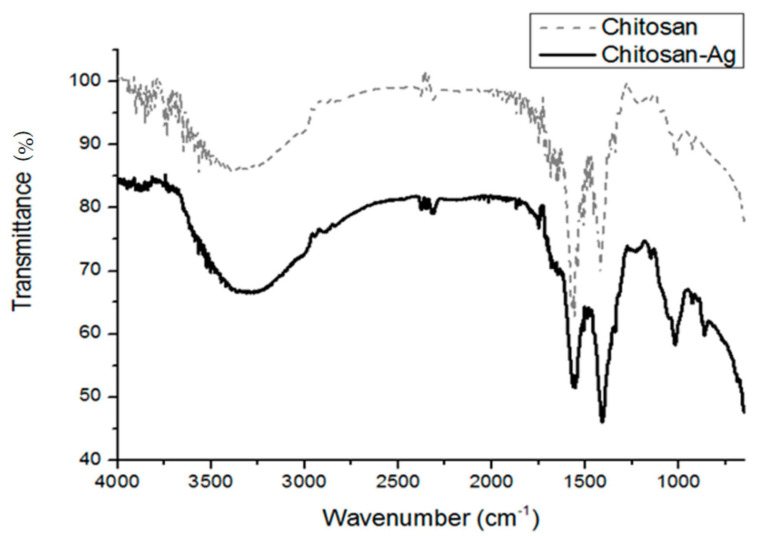
FTIR spectra of pure chitosan and chitosan/Ag membranes.

**Figure 8 polymers-14-01764-f008:**
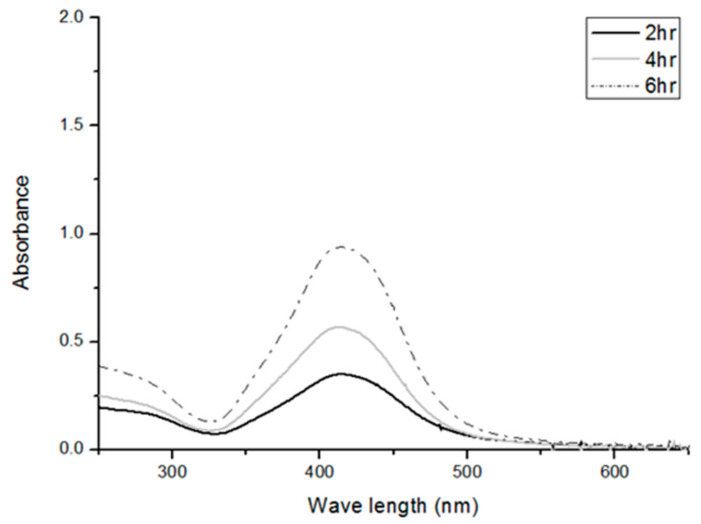
UV-Vis absorbent spectra of Ag particles as related to reaction time (2, 4, and 6 h) and incident UV-Vis wavelength (250–650 mm).

**Figure 9 polymers-14-01764-f009:**
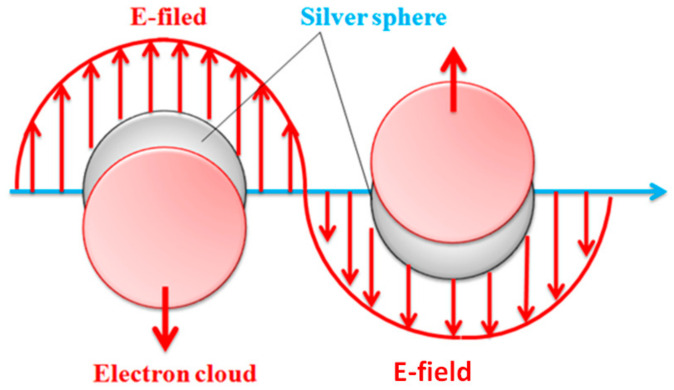
Nanoparticles in plasmon oscillation.

**Figure 10 polymers-14-01764-f010:**
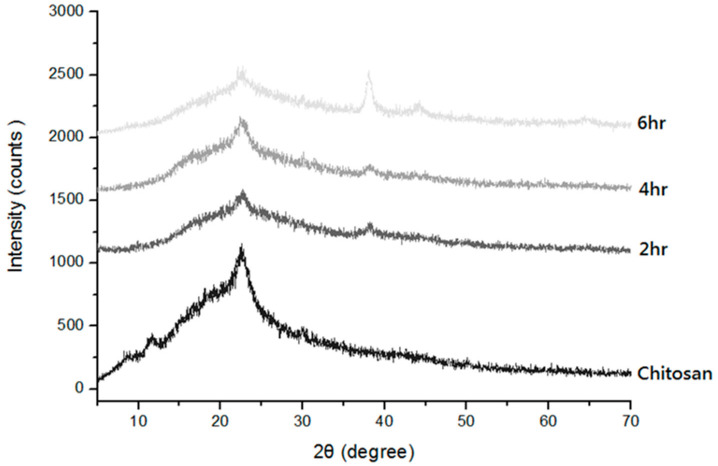
X-ray diffraction spectrum of chitosan/Ag membranes as related to reaction time.

**Figure 11 polymers-14-01764-f011:**
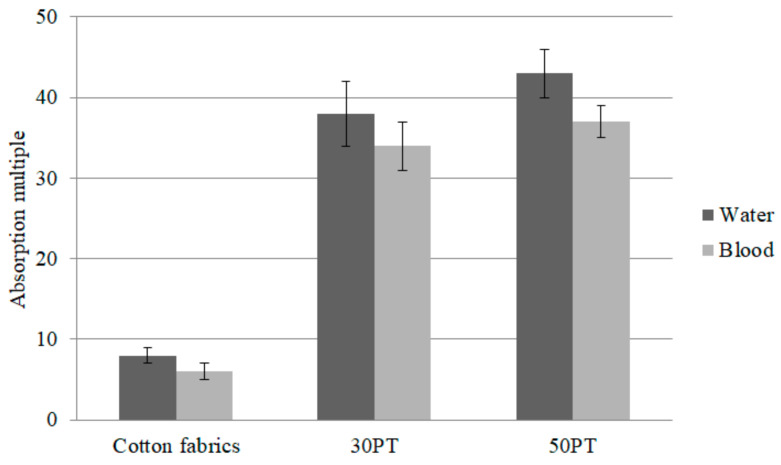
Moisture and blood absorption performances of polyacrylate/Tencel^®^ nonwoven composites as related to the content of polyacrylate fibers.

**Figure 12 polymers-14-01764-f012:**
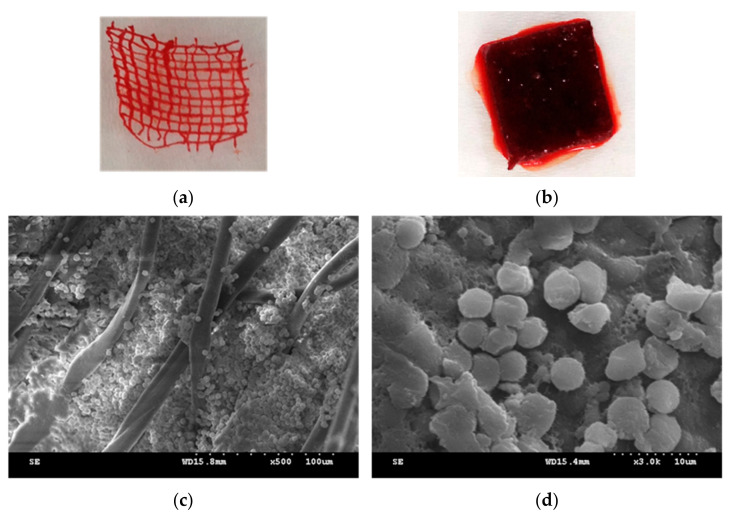
Blood absorption SEM images of (**a**) cotton fabrics and (**b**) 50PT nonwoven composites; (**c**,**d**) show 500× and 3000× magnification of 50PT nonwoven composites.

**Figure 13 polymers-14-01764-f013:**
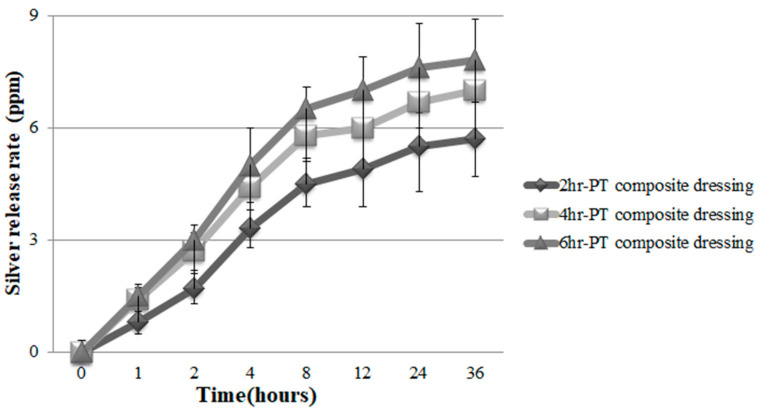
Silver release rate of PT nonwoven dressing with various reaction time.

**Figure 14 polymers-14-01764-f014:**
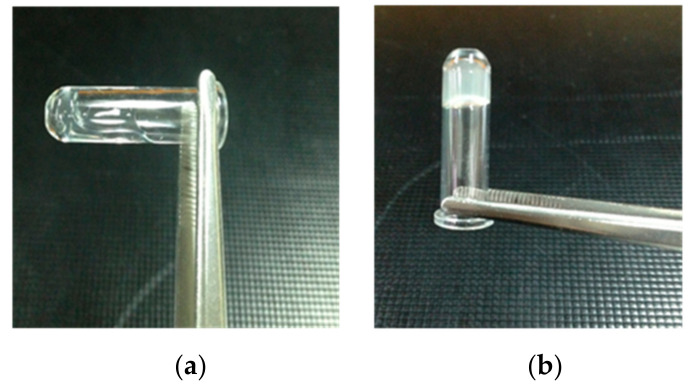
Derived from blood separation, plasma exhibits a fluid and a static states (**a**) before and (**b**) after an agglutination reaction, respectively.

**Figure 15 polymers-14-01764-f015:**
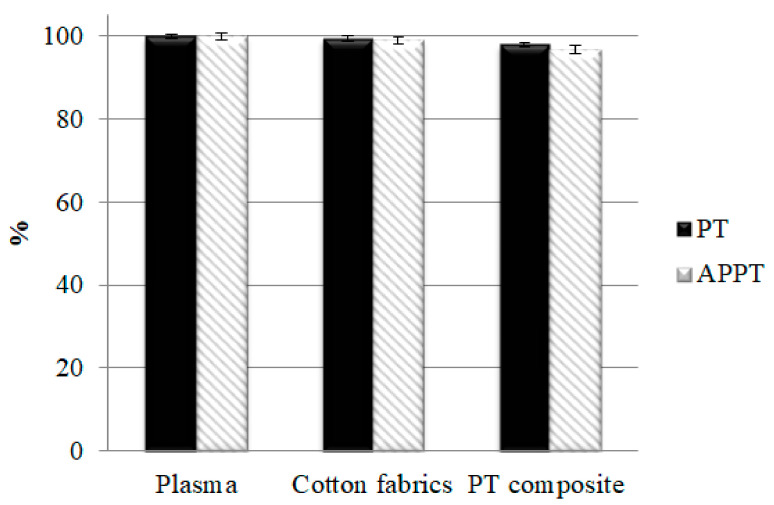
PT and APTT of polyacrylate/Tencel^®^ nonwoven composite.

**Figure 16 polymers-14-01764-f016:**
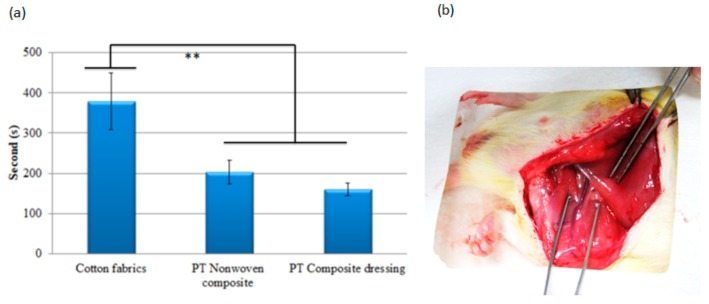
(**a**) Hemostasis time of cotton fabrics, PT nonwoven composite, and PT composite dressing in animal model, ** *p* < 0.01; (**b**) Hemorrhage from the cut in a rat’s femur artery.

**Table 1 polymers-14-01764-t001:** Denotations of PT nonwoven composites used for blood absorption measurement.

Nonwoven Composite	Polyacrylate Fiber (wt %)	Tencel^®^ (wt %)	Area Weight (g/m^2^)	Punching Density (Punches/cm^2^)	Thickness (mm)
10 PT	10	90	100	100	0.8
20 PT	20	80	100	100	0.8
30 PT	30	70	100	100	0.8
40 PT	40	60	100	100	0.8
50 PT	50	50	100	100	0.8

**Table 2 polymers-14-01764-t002:** The antibacterial property of PT nonwoven dressing with different reaction times.

	*Escherichia coli* (mm)	*Staphylococcus aureus* (mm)
PT nonwoven composite	0 ± 0	0 ± 0
2 h PT composite dressing	0.9 ± 0.06	0.2 ± 0.01
4 h PT composite dressing	1.2 ± 0.10	0.4 ± 0.04
6 h PT composite dressing	1.9 ± 0.09	0.8 ± 0.05

## Data Availability

All data relevant to the study are included in the article.

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
