# Peer review of "Hemostasis Evaluation of Antibacterial and Highly Absorbent Composite Wound Dressings in Animal Hemostasis Models"

_polymers, 2022, doi:10.3390/polym14091764_

Round 1

Reviewer 1 Report

Dear author, please revise your manuscript to the following suggested points

I strongly recommend revising this manuscript as follows:

  1. I would like to suggest to the authors, in the introduction paragraph line number 80 Chitosan and cellulose share comparable molecular structures where molecules are in the form of straight chains. In addition to strong polarity and ease of crystallization, chitosan has a melting point that is higher than the self-decomposition temperature, which makes it difficult to acquire amorphous chitosan. In medical applications, chitosan helps accelerate platelet aggregation and has low toxicity, biodegradable property, and good biocompatibility. The author should cite the following very recent artcles.https://doi.org/10.1002/mabi.202000395and https://doi.org/10.1016/j.carbpol.2022.119289
  2. The author should add a schematic illustration of Nonwoven Composites preparation and must be added the image of Nonwoven Composites in this illustration that will be very helpful to the readers.
  3. The author should add the in-vitro data on the dissolution rate of the Nonwoven Composites in the revised manuscript.
  4. Authors should revise figure no 11 b with better quality of the image.
  5. The author should be added the control data in each experimental result of PT, APTT, and Platelet Agglutination Analyses of PT Nonwoven Composites.
  6. The author should include the information ( a separate graph) about the Ag release rate from the Nonwoven Composites.

Author Response

Reviewer 1

Comments and Suggestions for Authors

Dear author, please revise your manuscript to the following suggested points

I strongly recommend revising this manuscript as follows:

  1. I would like to suggest to the authors, in the introduction paragraph line number 80 Chitosan and cellulose share comparable molecular structures where molecules are in the form of straight chains. In addition to strong polarity and ease of crystallization, chitosan has a melting point that is higher than the self-decomposition temperature, which makes it difficult to acquire amorphous chitosan. In medical applications, chitosan helps accelerate platelet aggregation and has low toxicity, biodegradable property, and good biocompatibility. The author should cite the following very recent artcles.https://doi.org/10.1002/mabi.202000395and https://doi.org/10.1016/j.carbpol.2022.119289

Ans: Thank you for your comments. The reference has been added in the introduction.

  1. The author should add a schematic illustration of Nonwoven Composites preparation and must be added the image of Nonwoven Composites in this illustration that will be very helpful to the readers.

Ans: Thank you for your comments. The schematic illustration of nonwoven composites was added to the Materials and Methods.

  1. The author should add the in-vitro data on the dissolution rate of the Nonwoven Composites in the revised manuscript.

Ans: Thank you for your comments. Combining Polyacrylate fibers and Tencel® fibers, the proposed nonwoven composites possess excellent dimensional stability and also retain a full structural integrity after absorbing moisture or blood. In addition to the dissolution test, the moisture absorption and Ag release rate of samples have been incorporated with the text.

  1. Authors should revise figure no 11 b with better quality of the image.

Ans: Thank you for your comments. Figure 11 (b) has been replaced with image that has better resolution.

  1. The author should be added the control data in each experimental result of PT, APTT, and Platelet Agglutination Analyses of PT Nonwoven Composites.

Ans: Thank you for your comments. We have added the control data in each experimental result of PT, APTT, and Platelet Agglutination Analyses of PT nonwoven composites.

  1. The author should include the information ( a separate graph) about the Ag release rate from the Nonwoven Composites.

Ans: Thank you for your comments. Ag release rate of nonwoven composites has been appended to the test results.

Reviewer 2 Report

The authors prepared Tencel® fibers and polyacrylate fibers into polyacrylate /Tencel® (PT) nonwoven composites. composites were evaluated in terms of physical evaluations as related to the fiber blending ratio, thereby acquiring the optimal moist absorption for PT nonwoven composites. The work is well presented and the results are interesting. However, I have some issues that should be addressed before acceptance:

-Some parts of paper were written with different color. Please correct

-please added antibacterial to Keywords

-in abstract, please give a clear conclusion from your work

-ther are some new papers on nanofibers and biomedical uses, please cite following articles to enrich your manuscript and compare with them: https://pubmed.ncbi.nlm.nih.gov/35269272/, https://doi.org/10.1016/j.electacta.2022.139892, https://doi.org/10.1016/j.ajps.2020.05.003, https://doi.org/10.1016/j.jddst.2021.102428 and https://doi.org/10.3390/polym14061259

-there are some grammar errors and mispronunciation In the whole text, for example page 2 line 93 “acquirign” , “optial” and more.

-Figures 1 and 3 have low quality, please increase their resolution

-The authors used SEM analysis but there is not any of it in “methods” section. Detailed methodology is expected.

-why didn’t bring the XRD of Ag NPs alone?

-How do you conclude that Ag NPs were at nanometric range with out SEM, DLS or XRD crystalline size? Please explain.

Author Response

Reviewer 2

Comments and Suggestions for Authors

The authors prepared Tencel® fibers and polyacrylate fibers into polyacrylate /Tencel® (PT) nonwoven composites. composites were evaluated in terms of physical evaluations as related to the fiber blending ratio, thereby acquiring the optimal moist absorption for PT nonwoven composites. The work is well presented and the results are interesting. However, I have some issues that should be addressed before acceptance:

 1 Some parts of paper were written with different color. Please correct

Ans: Thank you for your comments. The different color indicates the corrections or supplementations.

2 please added antibacterial to Keywords

Ans: Thank you for your comments. The antibacterial was added in keywords.

3 in abstract, please give a clear conclusion from your work

Ans: Thank you for your comments. A clear conclusion was supplemented with the abstract.

4 ther are some new papers on nanofibers and biomedical uses, please cite following articles to enrich your manuscript and compare with them: https://pubmed.ncbi.nlm.nih.gov/35269272/, https://doi.org/10.1016/j.electacta.2022.139892, https://doi.org/10.1016/j.ajps.2020.05.003, https://doi.org/10.1016/j.jddst.2021.102428 and https://doi.org/10.3390/polym14061259

Ans: Thank you for your comments. More literatures related to nanofibers and biomedical properties have now been provided in the introduction.

5 there are some grammar errors and mispronunciation In the whole text, for example page 2 line 93 “acquirign” , “optial” and more.

Ans: Thank you for your comments. The text has now been reviewed and corrected accordingly.

6 Figures 1 and 3 have low quality, please increase their resolution

Ans: Thank you for your comments. The two figures were revised with highly improved resolution.

7 The authors used SEM analysis but there is not any of it in “methods” section. Detailed methodology is expected.

Ans: Thank you for your comments. The introduction of SEM analysis was added to the methods.

8 why didn’t bring the XRD of Ag NPs alone?

Ans: Thank you for your comments. Ag nanoparticles alone cannot be directly analyzed through the X-ray diffraction spectrum because Ag particles are formed as a result of the blending with chitosan in a specified length of time (i.e. 2, 4, or 6 hours).

9 How do you conclude that Ag NPs were at nanometric range with out SEM, DLS or XRD crystalline size? Please explain.

Ans: Thank you for your comments. The formation of Ag nanoparticles is confirmed by the results of the UV-Vis absorbent spectra and the X-ray diffraction spectrum. Moreover, Ag nanoparticles was proven to be at a nano grade in our previous study by means of the observation with a transmission electron microscope.

Round 2

Reviewer 2 Report

the authors addressed some of my comments. but, it needs to add suggested references as I mentioned before. these references give readers an insight into similar studies in this field to compare related works.